# *APOL1* genotype associated risk for preeclampsia in African populations: Rationale and protocol design for studies in women of African ancestry in resource limited settings

**Charlotte Osafo**[1,2]*, **Nicholas Ekow Thomford**[3,4], **Jerry Coleman**[5‡], **Abraham Carboo**[6‡], **Chris Guure**[7‡], **Perditer Okyere**[7‡], **Dwomoa Adu**[1☉], **Richard Adanu**[8☉], **Rulan S. Parekh**[9☉], **David Burke**[10☉]

1 Department of Medicine and Therapeutics, School of Medicine and Dentistry, College of Health Sciences, University of Ghana, Accra, Ghana, 2 The Bank Hospital, Cantonment, Accra, Ghana, 3 Pharmacogenomics and Genomic Medicine Group, Department of Medical Biochemistry, School of Medical Sciences, College of Health and Allied Sciences, University of Cape Coast, Cape Coast, Ghana, 4 Division of Human Genetics, Department of Pathology, Faculty of Health Sciences, University of Cape Town, Cape Town, South Africa, 5 Department of Obstetrics and Gynecology, Korle Bu Teaching Hospital, Accra, Ghana, 6 School of Public Health, College of Health Sciences, University of Ghana, Accra, Ghana, 7 School of Medical Sciences, KNUST, Kumasi, Ghana, 8 Ghana College of Physicians and Surgeon, Accra, Ghana, 9 Departments of Pediatrics and Medicine, Hospital for Sick Children, University of Health Network, University of Toronto, Toronto, Canada, 10 Department of Human Genetics, University of Michigan Medical School, Ann Arbor, Michigan, United States of America

☉ These authors contributed equally to this work.
‡ JC, AC, CG and PO also contributed equally to this work.
* charlotte.osafo@thebankhospital.com

**Data Availability Statement:** All relevant data are within the paper.

## Abstract

### Background

Women of African ancestry are highly predisposed to preeclampsia which continues to be a major cause of maternal death in Africa. Common variants in the *APOL1* gene are potent risk factor for a spectrum of kidney disease. Recent studies have shown that *APOL1* risk variants contribute to the risk of preeclampsia. The aim of the study is to understand the contribution of *APOL1* risk variants to the development of preeclampsia in pregnant women in Ghana.

### Methods

The study is a case-control design which started recruitment in 2019 at the Korle Bu Teaching Hospital in Ghana. The study will recruit pregnant women with a target recruitment of 700 cases of preeclampsia and 700 normotensives. Clinical and demographic data of mother- baby dyad, with biospecimens including cord blood and placenta will be collected to assess clinical, biochemical and genetic markers of preeclampsia. The study protocol was approved by Korle Bu Teaching Hospital Institutional Review Board (Reference number: KBTH-IRB/000108/2018) on October 11, 2018.

**Funding:** CO- Fogarty International Center of the National Institutes of Health under Award Number K43TW011160. The funders had and will not have a role in study design, data collection and analysis, decision to publish, or preparation of the manuscript.

**Competing interests:** The authors have declared that no competing interests exist.

## Preliminary results

As of December 2021, a total of 773 mother-baby pairs had been recruited and majority of them had complete entry of data for analysis. The participants are made up of 384 pre-eclampsia cases and 389 normotensive mother-baby dyad. The mean age of participants is 30.69 ± 0.32 years for cases and 29.95 ± 0.32 for controls. Majority (85%) of the participants are between 20-30years. At booking, majority of cases had normal blood pressure compared to the time of diagnosis where 85% had a systolic BP greater than 140mmHg and a corresponding 82% had diastolic pressure greater than 90mmHg.

## Conclusion

Our study will ultimately provide clinical, biochemical and genotypic data for risk stratification of preeclampsia and careful monitoring during pregnancy to improve clinical management and outcomes.

## Introduction

Up to 8% of pregnancies are complicated by hypertensive disorders [1] resulting in maternal and neonatal morbidity and mortality particularly in developing countries [1, 2]. Pregnancy-related hypertensive disorder is referred to as preeclampsia which is characterized by new onset of hypertension (blood pressure ≥140/90mmHg) and proteinuria (greater than 1+ or 300mg per 24hours) after 20 weeks gestation [3, 4]. This definition has been expanded to include renal, liver, hematological and neurological dysfunction and fetal growth restriction [5]. The prevalence of preeclampsia in sub-Saharan Africa is high with an incidence rate of over 15% [6]. In sub-Saharan regions such as West Africa and especially Ghana, there is a reported incidence of approximately 8% [7]. A systematic analysis of prevalence of preeclampsia across sub-Saharan Africa has seen an increase over the past decade [6]. Women of African ancestry are highly predisposed to preeclampsia with a greater propensity to develop into a more complicated form known as HELLP (H = Haemolysis, EL = Elevated Liver enzymes, LP = Low Platelets) syndrome [8–11]. It is estimated that the prevalence of preeclampsia is higher in indigenous African women and those of African ancestry diaspora [8] than any other racial groups. This susceptibility to preeclampsia appears independent of socioeconomic status and may potentially be due to genetic factors [12, 13].

Genetic susceptibility to preeclampsia has for several decades been suspected but there is very limited research in this area especially among women of African ancestry. Predominant functional candidate genes studies in pre-eclampsia have targeted mechanisms such as thrombophilia (*F5*, *MTHFR*, *F2*, *SERPINE1*), Endothelial function (*eNOS3*, *VEGFR1*), vasoactive proteins (*AGT*, *ACE*), oxidative stress and lipid metabolism (*APOE*, *EPHX*, *GST*) and immunogenetics (*TNF*, *IL10*). Such studies have mostly been conducted in other populations and often target only maternal genotype.

Despite the long-term sequelae of preeclampsia, which includes increased risk of hypertension, cardiovascular disease, chronic kidney disease (CKD) and end stage renal disease (ESRD) among women of African descent, little is known about genetic risk factors such as apolipoprotein L1 (*APOL1)* risk. Common variants in the apolipoprotein L-1 gene (*APOL1*) only found predominantly in West Africans, make up the 2 risk haplotypes termed G1 and G2, which are potent risk factors for a spectrum of kidney diseases [14] but very rare in European

ancestry individuals. There is a strong relationship between kidney function and hypertension [15, 16] and considering the importance of the *APOL1* gene to people of African descent, an in-depth understanding of the role of this gene in preeclampsia in indigenous African women will be key to understanding preeclampsia genetics and advance biomarker innovations.

*APOL1* high risk variants [17] are present at high frequencies in populations of West African descent and account for much of increased risk of non-diabetic chronic kidney disease [18, 19]. Among populations from Nigeria and Ghana, the *APOL1* high-risk genotype frequency approaches 25% and 13% in African Americans who share west African ancestry. The frequencies are enriched among those with chronic kidney disease [18, 19]. It is known that preeclampsia results in part from microangiopathy in the glomerulus of the kidney suggesting a potentially important role of APOL1 in preeclampsia [20, 21]. Evidence from previous studies have found associations of *APOL1* risk variants with microangiopathy in HIV positive patients [22, 23]. In a transgenic mouse model, a pregnancy-associated phenotype that encompassed eclampsia, preeclampsia, fetal wasting occurred in some Tg-G0 mice and Tg-G2 mice. Placentas of Tg mice expressed APOL1, similar to human placenta, suggesting a role for APOL1 in preeclampsia [24]. Small sample sized studies have looked at the genotype associated risk for preeclampsia in women of African ancestry [25, 26] mostly looking at African Americans. *APOL1* conferred a 2-fold increased risk of preeclampsia for fetuses carrying homozygotes or compound heterozygotes (G1/G1, G2/G2, or G1/G2), termed the high-risk genotypes. In addition, high-risk *APOL1* genotypes may present a higher proportion in infants born from birth complications by preeclampsia and may have poor perinatal outcomes [27]. These studies thus suggest there is an interaction between the fetus and/or placenta resulting in causal effects for preeclampsia resulting in a potential long-term effect on CKD development in women [26]. It is also conceivable that APOL1 expression in the placenta may play a causal role in preeclampsia, which may be modulated by either maternal or infant *APOL1* genotype.

To date there has been no studies in indigenous women of African ancestry specifically focusing on APOL1 associated risk factors with a highly powered sample size. Most studies have been in African Americans without sufficiently large sample size, which has been mostly the challenge. Developing a study that is aimed specifically at genotype associated risk preeclampsia allows for accurate biomedical and clinical phenotyping and identification of genotype-phenotype associated risk factors with clinical and birth outcomes. There still remains a gap in differential impact of *APOL1* status of mother and child with either maternal or perinatal outcomes. This study would give some idea about the penetrance of *APOL1* in African populations experiencing different demographics and environmental stressors than African descendants living in the USA. The overarching objective for the *APOL1* characterization in preeclampsia is to undertake a comprehensive genotype-phenotype of a cohort with translational potential. This study involves a team of clinicians with various specialties, scientists and has capacity building component as part of the study.

## Aims and objectives

The overall aim is to understand the influence of *APOL1* on preeclampsia and its sequelae on both perinatal and maternal outcomes using a case- control design. The study will: (1) examine the association of *APOL1* risk variants in pregnant women with preeclampsia compared to normotensive pregnant women, (2) determine the risk of perinatal outcomes among infants from mothers with pre-eclampsia compared to infants from normotensive mothers by *APOL1* risk variant status and (3) Undertake a longitudinal follow up of both the preeclamptic and normotensive mothers recruited into the study to determine the incidence of non-communicable

> ## Box 1. Impact and innovation of study
>
> - Determination of risk of both maternal and infant APOL1 risk variants and association with preeclampsia.
>
> - Have novel insights into genetic factors that increase risk for worse perinatal outcomes.
>
> - Create a research platform with a well characterized cohort and specimen repository for clinical and translational studies in preeclampsia among women of African ancestry for African investigators.
>
> - Harness a simple genotyping platform to develop a preconception screening platform for high-risk women who present at a health facility.
>
> - Developing genetic tools to predict risk of preeclampsia in women of African ancestry with direct impact on the clinical care during pregnancy and perinatal outcomes, as well as reducing maternal fatality.

diseases including Chronic Kidney Disease (CKD), Hypertension, Stroke, Mental Health and Cardiovascular diseases among the women with preeclampsia versus normotensive women stratified by *APOL1* status.

## Impact and innovation

This study is key to addressing an important disease related to pregnancy with a potential for biomarker innovation. One unique aspect of this study is the population cohort of black African women. Outcome from the study will provide data on risk stratification for preeclampsia and information on careful monitoring during pregnancy with clinical correlation (Box 1).

This study is timely and addresses an important public health issue as maternal health improves the health and welfare of entire communities.

## Methods

### Study design/population

The work describes a case-control study design with a target number of 700 cases of preeclampsia and 700 cases of normotensive pregnant women, prospectively enrolled. Recruitment started in May 2019 involving women who are diagnosed with preeclampsia for the very first time as cases. The target population are African women; however, the only recruiting facility is KBTH in Ghana. The work will capture a significant number of Ghanaian women with longitudinal follow-up over 3 years. All pregnant women will be eligible for recruitment. Eligible women will be screened and enrolled into the study after provision of informed consent.

### Ethical consideration

The study protocol was approved by the Korle Bu Teaching Hospital Scientific and Technical committee/ Institutional Review Board (reference number: KBTH-STC/IRB/000108/2018).

**Table 1. Inclusion and exclusion criteria.**

| Inclusion criteria | Exclusion criteria |
|---|---|
| **Cases** | |
| Preeclampsia will be defined as women with new onset of hypertension (blood pressure ≥140/90mmHg) and proteinuria (greater than 1+ or 300mg per 24hours) after 20 weeks gestation | Prior history of preeclampsia |
| Patients with preeclampsia, who develop eclampsia and HELLP syndrome | Patients with prior chronic hypertension defined as hypertension that precedes pregnancy or occurs in the first half of pregnancy |
| **Controls** | Patients with existing co-morbid conditions such as diabetes, antepartum hemorrhage, other endocrine conditions or any secondary hypertension |
| Participants who did not have a diagnosis of preeclampsia at any time during the pregnancy | Patients who have had blood transfusion in the last three months |
| Patients who sign informed consent | Patients who do not sign informed consent |

Written informed consent was obtained from participants after explaining to them about the study with the option of discontinuing without any consequences. The inclusion and exclusion criteria are shown in Table 1.

## Overview of data collection

Participants involved in the study went through a comprehensive clinical assessment, an overview of which is shown in Fig 1. This assessment also included each baby that was born.

**Recruitment strategy.** The recruitment team which is made up of nurses, doctors and medical laboratory scientists underwent training and familiarization with the goals of the project at the maternity block of Korle Bu Teaching Hospital (KBTH), which is a university affiliated teaching hospital. The team was then trained on how to obtain informed consent, and how to use the Redcap app to collect data. In addition, the team was taken through a practical session on how to take cord blood and placenta sample, how to snap freeze placenta sample

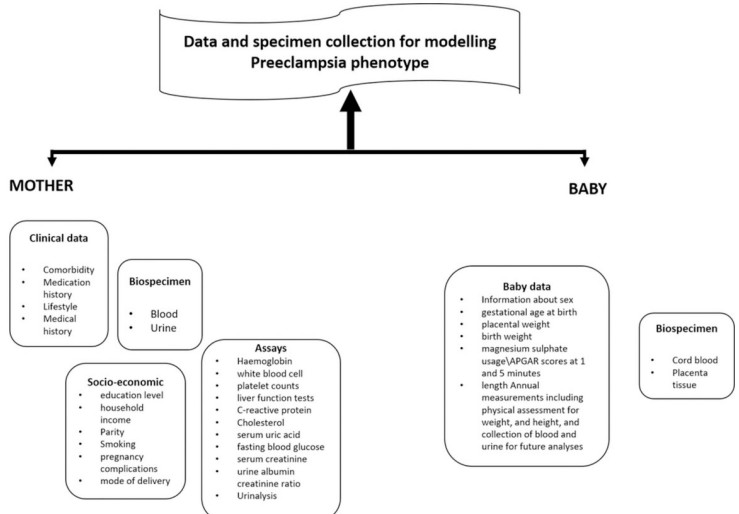

**Fig 1. Overview of the clinical assessment.**

and cord blood and how to package maternal samples for collection and processing by the laboratory technician.

To ensure efficient patient recruitment and improve communication among team members, a WhatsApp group was created for the team members to alert members of potential patients for recruitment and to report issues that needed urgent attention.

Our data collection instrument was transferred onto the redcap app which was initially implemented on three android tablets. After an initial pilot, we noticed that it was easier on phone so the app was installed on the phone. Each member of the recruitment team was given a unique identifier available for use from an earlier project that had been completed.

Patients were assessed as potential candidates for recruitment based on their medical history. The clinical team talked to the patient in the language the patients understood and then sought consent. Once consent was given, patients were recruited into the study, given a unique code and a comprehensive questionnaire administered. Clinical information was taken from patients' folders and then study visits scheduled based on patient's appointment at the facility.

Since study participants attended antenatal clinics, study related data collection was scheduled to coincide with the hospital visit so as not to inconvenience them. The schedules of all antenatal visits were maintained and the estimated time of delivery kept. On the day of delivery, samples were collected and newborn baby assessed.

**Data collection and quality control.** The target participants for this study include a minimum of 1400 participants of age-matched cases of patients with preeclampsia and controls of normotensive patients. The sample size has enough statistical power to allow a robust interpretation of the variables that will be collected as part of the cohort and clinical outcomes which will serve as our phenotypes. The data collected is collated and entered into Redcap by a team of clinicians, research scientists, laboratory technicians and data entry clerks. To maintain data validity and quality control, regular meetings were held with the study coordinator and recruitment team to review enrollment, data entry, missing information and specimen collection. The data are stored in a database that provides detailed recording of clinical, demographic and phenotyping characteristics of the cohort. Access to Redcap is limited to key personnel of the research team that ensures data quality and reconciliation.

**Sample collection and laboratory assays.** During the course of the study, we established an accompanying biorepository which will be available for future study. Samples collected from pregnant mothers included whole blood and urine on the day of recruitment. All samples collected are barcoded for both mother and child and stored at -80 degrees Celsius. At delivery, placenta and cord blood were collected. Placental samples were collected from a standardized location approximately 2cm beside the umbilical cord insertion, from the middle layer of placenta midway between maternal and fetal surfaces. The samples were cut into 1cm cubes and snap frozen in liquid nitrogen. Cord blood was snap frozen within 10mins of collection and then stored at -80 degree Celsius. Blood samples were processed by MDS laboratory. Laboratory tests were conducted according to local laboratory protocols. All biochemical tests were undertaken with automated analysers at the research lab of MDS Lancet laboratories, Ghana (https://www.cerbalancetafrica.com.gh/). All patients recruited into the study underwent an initial laboratory testing to obtain a baseline hematological and kidney function tests. For the mothers, hemoglobin, white blood cell and platelet counts, liver function tests, serum creatinine and urine albumin creatinine ratio were conducted. Several biomarkers have been used to diagnose preeclampsia and serve as valuable indicators. These include (i) renal impairment markers (serum creatinine, urine albumin creatinine ratio) (ii) liver dysfunction markers (AST, ALT) (iii) hypertension markers, and (iv) reduced platelets. These markers were used for phenotyping of preeclampsia and interpreted through the genetic model that will be

**Table 2. Biomarkers and time points for collection.**

| Preeclampsia | | Renal function | | Inflammation | |
|---|---|---|---|---|---|
| Biomarker | Time Point for Collection | Biomarker | Time Point for Collection | Biomarker | Time Point for Collection |
| VEGF | At recruitment | SCr | At recruitment and annually | CRP | At recruitment |
| PIGF | | UACR | | IL-6 | |
| s-FLT-1 | | - | | IL-8 | |
| s-Eng | | - | | TNF-alpha | |

VEGF: Vascular Endothelial Growth Factor; PIGF: Placenta Growth Factor; s-FLT-1: Soluble fms-like tyrosine kinase; s-Eng: Soluble Endoglin; sCr: Serum Creatinine; UACR: Urine Albumin Creatinine Ratio; CRP: C-reactive protein; IL-6: Interleukin 6; IL-8: Interleukin 8; TNF-alpha: Tumour Necrosis Factor alpha.

generated. Random duplicate samples will be taken for validation of laboratory measurements periodically. We plan to measure biomarkers for preeclampsia (Table 2), conduct placental histology and also follow up both babies and their mothers to determine the incidence of chronic kidney disease and other non-communicable diseases. Preeclampsia is not an easy condition to diagnose and thus physicians typically rely on several symptoms to guide diagnosis (Fig 2).

## Genetic analysis

DNA has been isolated and currently samples are being prepared for APOL1 genotyping. In this study multiple SNP analysis will be performed to further explore the genetic predisposition to preeclampsia in West African women. Selected *APOL1* SNPs to be considered are G1: rs73885319, rs60910145, and G2:rs71785313, rs12106505. Several studies on the genetic predisposition to preeclampsia have attempted to use candidate gene approach. These candidate gene approach have often focused on maternal genes as causative genes. Preeclampsia appears to be associated with *APOL1* variants *G1* and *G2* [28, 29]. However *APOL1* variants in preeclampsia are complex, as some studies have linked disease with maternal *APOL1* [30], while others with fetal *APOL1* variants [25, 27]. Genotyping will be undertaken in Ghana using predesigned TaqMan assays in the Pharmacogenomics and Genomic Medicine laboratory (www.pgmg-lab.com), School of Medical Sciences, University of Cape Coast.

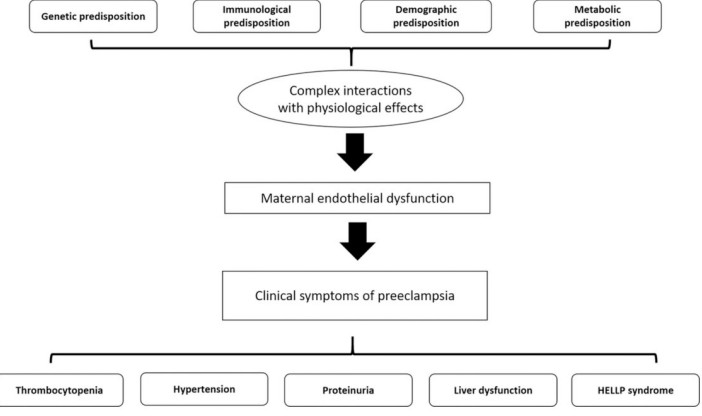

**Fig 2. Summarized pathogenesis of preeclampsia.**

## Current status of the study

### Preliminary data and current status of study

Recruitment is currently on going, with a target goal of 1400 participants. however as of 2021, a total of 773 mother baby dyad have been recruited making more than half of each recruitment group. These were made up of 384 pre-eclampsia individuals and 389 normotensive mother-baby dyad. Preliminary data and baseline characteristics of the participants are shown in Table 3.

The mean age of the preeclampsia patients is 30.69 ± 0.32 years, while that of normotensive controls is 29.95 ± 0.32 years. The majority (~85%) of the participants are between 20–30 years, with 70% of them being married. At recruitment, 82% of the preeclampsia participants were having normal systolic blood with a corresponding 77% diastolic blood pressure (Table 2). After 20 weeks 85% had systolic blood pressure of more than 140mmHg for pre-eclampsia participants with 82% having a diastolic blood pressure of greater than 90mmHg. Normotensive controls had more than 90% of the participants having a systolic blood pressure of less than 140mmHg and a diastolic blood pressure of less than 90mmHg.

Clinical and biochemical indicators of preeclampsia is shown in Table 4. The preliminary results indicate variations in the various biochemical parameter of susceptibility to preeclampsia and phenotyping.

Table 5 shows family history of diseases that could predispose mothers to preeclampsia. Majority of participants had no family history of diabetes mellitus, sickle cell disease, asthma, preeclampsia, birth defects or multiple pregnancies.

## Discussion and study challenges

Preeclampsia is a significant complication in pregnant women with consequences of morbidity and mortality for both the mother and the child. Despite several decades of research and an etiology leading to disease, clinicians are unable to predict and manage preeclampsia prior to symptom onset. Traditionally, clinicians have relied on maternal risk factors such as age, family history and comorbidities in trying to identify "at-risk" women. These traditional risk factors are generalized and non-modifiable in several conditions and thus affects accuracy in classifying women. Several angiogenic biomarkers are being developed in several populations to improve prediction [30–32].

Though angiogenic markers may be helpful in diagnosis and eventual management of pre-eclampsia, identifying genetic biomarkers in preeclampsia will increase the prediction of genetic predisposition and associated triggers and environments. Genetic predisposition to preeclampsia plays a significant role in the condition and identifying "at risk" women may prevent preeclampsia-related complications. The gene *APOL1* has been implicated in pre-eclampsia, consequently, the role of APOL1 in populations of African origin may achieve opportunities for improved diagnosis and management. Given the utility of genetic screening in medical genetics practice, the uniqueness and potential for *APOL1* screening to identify "at risk" women will be very useful. The current study has a recruitment target of 1400 mother-baby dyad in a longitudinal study design that includes comprehensive clinical assessments at sequential pre-birth timepoints and post-delivery clinical and morphologic assessment. There is a comprehensive plan to follow up on both mother and baby to observe for any complications arising from preeclampsia during birth.

Our preliminary data so far have shown distinct characteristics among the preeclampsia patients and controls. During antenatal visits, an increase in blood pressure of ≥140/90mmHg after 20 weeks were observed in our preeclampsia patients. Biochemical and clinical parameters of preeclampsia were also observed in several of diagnosed preeclampsia patients. The majority

**Table 3. Characteristic of recruited patients with and without preeclampsia.**

| Variable | | Enrolled pairs | |
|---|---|---|---|
| | | Cases n (%) | Control n (%) |
| **N** | 773 | **384** | **389** |
| **Maternal age (years)** | | | |
| | *mean ± SE* | 30.69 ± 0.32 | 29.95 ± 0.32 |
| | *10–19* | 12 (3.13) | 29 (7.46) |
| | *20–29* | 167 (43.49) | 142 (36.50) |
| | *30–39* | 176 (45.83) | 194 (49.87) |
| | *40–49* | 28 (7.29) | 23 (5.91) |
| | *Missing* | 1 (0.26) | 1 (0.26) |
| **Marital status** | | | |
| | *Married* | 278 (72.40) | 274 (70.44) |
| | *Single* | 92 (23.96) | 110 (28.28) |
| | *Co-habiting* | 7 (1.82) | 1 (0.26) |
| | *Missing* | 7 (1.82) | 3 (0.77) |
| **Maternal BMI (kg/m$^2$)** | | | |
| **@booking** | <18.5 | 22 (5.73) | 14 (3.60) |
| | 18.5–24.9 | 73 (19.01) | 121 (31.11) |
| | 25.0–29.9 | 108 (28.13) | 108 (27.76) |
| | 30–39.9 | 92 (23.96) | 79 (20.31) |
| | ≥40 | 16 (4.17) | 10 (2.57) |
| | *Missing* | 73 (19.01) | 57 (14.65) |
| **@20 weeks** | | | |
| | <18.5 | 0 (0) | 0 (0) |
| | 18.5–24.9 | 0 (0) | 1 (0.26) |
| | 25.0–29.9 | 1 (0.26) | 0 (0) |
| | 30–39.9 | 0 (0) | 0 (0) |
| | ≥40 | 175 (45.57) | 156 (40.10) |
| | *Missing* | 208 (54.17) | 232 (59.64) |
| **@ after 30weeks** | | | |
| | <18.5 | 112 (29.17) | 97 (24.94) |
| | 18.5–24.9 | 13 (3.39) | 28 (7.20) |
| | 25.0–29.9 | 46 (11.98) | 83 (21.34) |
| | 30–39.9 | 114 (29.69) | 106 (27.25) |
| | ≥40 | | |
| | *Missing* | 73 (19.01) | 57 (14.65) |
| **Ethnicity** | | | |
| | *Akan* | 180 (46.88) | 173 (44.47) |
| | *Ewe* | 42 (10.94) | 43 (11.05) |
| | *Ga* | 70 (18.23) | 95 (24.42) |
| | *Hausa* | 31 (8.07) | 24 (6.17) |
| | *Ga-Adangbe* | 9 (2.34) | 3 (0.77) |
| | *Others* | 31 (8.07) | 30 (7.71) |
| | *Missing* | 21 (5.47) | 21 (5.40) |
| **Education** | | | |
| | *No formal education* | 25 (6.51) | 18 (4.63) |
| | *Primary* | 35 (9.11) | 55 (14.14) |
| | *Junior High School* | 133 (34.64) | 103 (26.48) |
| | *Senior High School* | 109 (28.39) | 118 (30.33) |
| | *Vocational School* | 0 (0.00) | 3 (0.77) |
| | *Tertiary* | 60 (15.63) | 77 (19.79) |
| | *Missing* | 19 (4.95) | 1 (0.26) |

*(Continued)*

**Table 3.** (Continued)

| Variable | | Enrolled pairs | |
|---|---|---|---|
| | | Cases n (%) | Control n (%) |
| **Alcohol usage** | | | |
| | *Gave up before pregnancy* | 33 (8.59) | 24 (6.17) |
| | *Gave during Pregnancy* | 13 (3.39) | 8 (2.06) |
| | *Never* | 334 (86.98) | 355 (91.26) |
| | *Missing* | 4 (1.04) | 2 (0.51) |
| **Systolic BP (mmHg)** | | | |
| **@booking** | *60–139* | 315 (82.03) | 362 (93.06) |
| | *>140* | 41 (10.68) | 2 (0.51) |
| | missing | 28 (7.2) | 25 (6.43) |
| **@diagnosis** | *60–139* | 42 (10.94) | 358 (92.03) |
| | *>140* | 332 (85.35) | 22 (5.66) |
| | missing | 10 (2.60) | 9 (2.31) |
| **Diastolic BP (mmHg)** | | | |
| **@booking** | *10–89* | 296 (77.08) | 352 (90.49) |
| | *>90* | 60 (15.63) | 10 (2.57) |
| | *Missing* | 28 (7.29) | 27 (6.94) |
| **@diagnosis**. | | | |
| | *10–89* | 54 (14.06) | 347 (89.20) |
| | *>90* | 318 (82.81) | 33(8.48) |
| | *Missing* | 12 (3.13) | 9 (2.31) |
| **Gravidity** | | | |
| | *1* | 91 (23.70) | 87 (22.37) |
| | *2* | 77 (20.05) | 1 (0.26) |
| | *3* | 61 (15.89) | 1 (0.26) |
| | *4* | 57 (14.84) | 1 (0.26) |
| | *5* | 45 (11.72) | 102 (26.22) |
| | *6* | 19 (4.95) | 74 (19.02) |
| | *7* | 19 (4.95) | 61 (15.68) |
| | *8* | 4 (1.04) | 34 (8.74) |
| | *9* | 4 (1.04) | 15 (8.74) |
| | *10* | 4 (1.04) | 5 (1.29) |
| | *Missing* | 3 (0.78) | 2 (0.51) |

of patients had no family history of disease, preexisting hypertension, sickle cell disease, or prior preeclampsia. Our methodology and recruitment strategies have been designed with an aim to addressing and standardizing evidential gaps especially in phenotyping. Data collection has been comprehensive and 1033 data collection tools capturing clinical and treatment history, biochemical markers and clinical features of both mother and babies have been obtained.

There is a limitation to our study. Our sample was limited to women without a prior history of preeclampsia. This removes from our study population women who might have recurrent preeclampsia due to their (or their fetus') APOL1 status. This may result in an underestimate of the relationship between APOL1 and preeclampsia.

## Challenges

**Protocol, recruitment, and patient attrition.** There have been several challenges during the implementation of the project. A primary challenge is the attrition rate of recruitment

**Table 4. Clinical parameters and hematological indicators of preeclampsia.**

| Variable | | Enrolled pairs | |
| --- | --- | --- | --- |
| | | Cases n (%) | Control n (%) |
| N | 773 | 384 | 389 |
| **Hb (g/dL)** | *0–11* | 212 (54.95) | 236 (60.67) |
| | *12–16* | 123 (32.03) | 103 (26.48) |
| | *Missing* | 49 (12.76) | 50 (12.85) |
| **PLT (x 10$^9$/L** | *5–150* | 46 (11.98) | 22 (5.66) |
| | *151–450* | 135 (34.70) | 159 (40.87) |
| | *451–600* | 1 (0.26) | 0 (0.00) |
| | *Missing* | 202 (52.60) | 208 (53.47) |
| **eGFR (ml/min/1.73m$^2$)** | *0–15* | 1 (0.26) | 0 (0.00) |
| | *16–29* | 5 (1.30) | 1 (0.26) |
| | *30–59* | 18 (4.69) | 2 (0.51) |
| | *60–89* | 199 (51.82) | 179 (46.02) |
| | *Missing* | 161 (41.93) | 207 (53.21) |
| **HCT (L/L)** | *0–0.47* | 205 (53.39) | 194 (49.87) |
| | *>0.47* | 1 (0.26) | 0 (0.00) |
| | *Missing* | 178 (46.35) | 195 (50.13) |
| **UACR (mg/mmol)** | *<3* | 10 (2.60) | 51 (13.11) |
| | *3–30* | 34 (8.85) | 101 (25.96) |
| | *>30* | 160 (41.67) | 40 (10.28) |
| | *Missing* | 180 (46.88) | 197 (50.64) |

nurses from the project in pursuit of other ventures. At the commencement of the project, we had several meetings and training activities with the nursing staff to map out strategies and the recruitment plan. Recruitment began appropriately, with approximately 30 patients (cases and controls) per month. However, as the trained nurses left, recruitment dropped while the replacement nurses were recruited and trained to tasks. Multiple cases and fetal samples were missed by newly-trained staff. Delivery times at late night and early mornings resulted in missing samples. Frequently, patients are in labor for several hours. During late evenings, trained team recruitment nurses complete their shifts and hand over to a colleague. Even with clear instructions, the take-over nurses fail to obtain samples. We solved this issue by employing a recruitment nurse during the night shift to assist in night recruitment and to reduce sample collection losses.

Table 6 summarizes the major challenges encountered in the project and interventions implemented and Fig 3 shows the change in cord sample numbers per year after intervention. The total percentage of cord blood and placental samples missed in 2021 was 8.41% compared with 17.92% in 2020.

**Data entry and integrity.** Our data is collected using the web-based Research Electronic Data Capture (REDCap) management system. Data quality was captured using a standard data entry interface. Records are updated regularly and the integrity checked weekly. Data entry has been allocated to specific data collectors and scientists to ensure accuracy. However, some missing data can be observed in the preliminary data shown in Tables 2–4. Missing data is primarily the result of: (1) missing research IDs for patients, (2) biochemical data lost at the testing laboratory, and (3) loss of Internet connectivity from field data collectors.

**Table 5. Family History of risk factors for participants with and without preeclampsia.**

| Variable | | Enrolled pairs | |
|---|---|---|---|
| | | Cases n (%) | Control n (%) |
| N | 773 | 384 | 389 |
| **Diabetes mellitus** | *No* | 362 (94.27) | 365 (93.83) |
| | *Yes* | 12 (3.13) | 17 (4.37) |
| | *Unknown* | 4 (1.04) | 3 (0.77) |
| | *Missing* | 6 (1.56) | 4 (1.03) |
| **Sickle Cell Disease** | *No* | 359 (92.29) | 363 (93.32) |
| | *Yes* | 12 (3.13) | 16 (4.11) |
| | *Unknown* | 8 (2.08) | 6 (1.54) |
| | *Missing* | 5 (1.30) | 4 (1.03) |
| **Asthma** | *No* | *363 (94.53)* | 368 (94.60) |
| | *Yes* | *9 (2.34)* | 6 (1.54) |
| | *Unknown* | *5 (1.30)* | 8 (2.06) |
| | *Missing* | 7 (1.82) | 7 (1.80) |
| **preeclampsia** | *No* | 361 (94.01) | 361 (92.80) |
| | *Yes* | 2 (0.52) | 1 (0.25) |
| | *Unknown* | 17 (4.43) | 18 (4.46) |
| | *Missing* | 4 (1.04) | 9 (2.31) |
| **Birth defects** | *No* | 374 (97.40) | 379 (97.43) |
| | *Yes* | 3 (0.78) | 4 (1.03) |
| | *Unknown* | 3 (0.78) | 3 (0.77) |
| | *Missing* | 4 (1.04) | 3 (0.77) |
| **Multiple pregnancies** | *No* | 272 (70.83) | 271 (69.67) |
| | *Yes* | 104 (27.08) | 111 (28.53) |
| | *Unknown* | 4 (1.04) | 3 (0.77) |
| | *Missing* | 4 (1.04) | 4 (1.03) |

**Genotyping and infrastructure development.** Preeclampsia and its associated health outcome phenotype measures must be supported by the acquiring of relevant genotype data. The global COVID-19 pandemic has disrupted local African research laboratory genotyping. Once a robust and consistent genotyping infrastructure is re-activated, we will complete the genotyping of these study samples. In tandem with the clinical research capacity demonstrated in this work, we continue to build local expertise in molecular genetics, clinical research skills, data analysis, and a biorepository for patient cohort samples.

## Conclusion

Our study addresses an important disease in pregnancy with significant morbidity and mortality that has not been well studied in large populations in Africans with a high burden of preeclampsia and *APOL1* high risk genotypes. Despite the difficulties in the recruitment of mother-baby dyad in a low middle income country as well as the sudden emergence of COVID-19 pandemic which brought recruitment to a standstill, our study has been successful in recruiting more than 700 participants within the last 26 months. The results obtained from our study will assist in developing genetic tools to predict risk of preeclampsia in women of African ancestry worldwide, with direct impact on the clinical care during pregnancy and perinatal outcomes as well as reducing maternal fatality.

**Table 6. Timelines of major challenges encountered and interventions implemented.**

| Challenge | Period of facing challenge/ implementation (months) | Effect of Challenge | Intervention | Effect of new intervention |
|---|---|---|---|---|
| Loss of trained recruitment nurses | 18 months | Affected recruitment numbers and missed samples | Increased the number of recruitment nurses per ward or floor and robust backup plans | Reduction in the loss of samples (Fig 3) |
| Parallel research projects competing for the same patients | 24 months | Reduction in recruitment numbers and obtaining enough blood volumes | Division of recruitment floors and patient management | Improvement in recruitment numbers and adequate blood volumes for further downstream analysis |
| Patients had difficult veins which meant we couldn't get enough blood volumes | 6 months | Inadequate blood volumes during recruitment | Use of vacutainer apparatus to increase blood volume | Volumes of blood increased |
| Hemolysis of blood samples | 6 months | Negatively affected biochemical analysis | Immediate separation of samples after collection | number of hemolysed samples reduced |
| Recruitment during early stages of pregnancy | 12 months | Loss to follow up and emergency deliveries | Recruitment was planned to reduce the time between recruitment and delivery | Reduction in loss of participants |

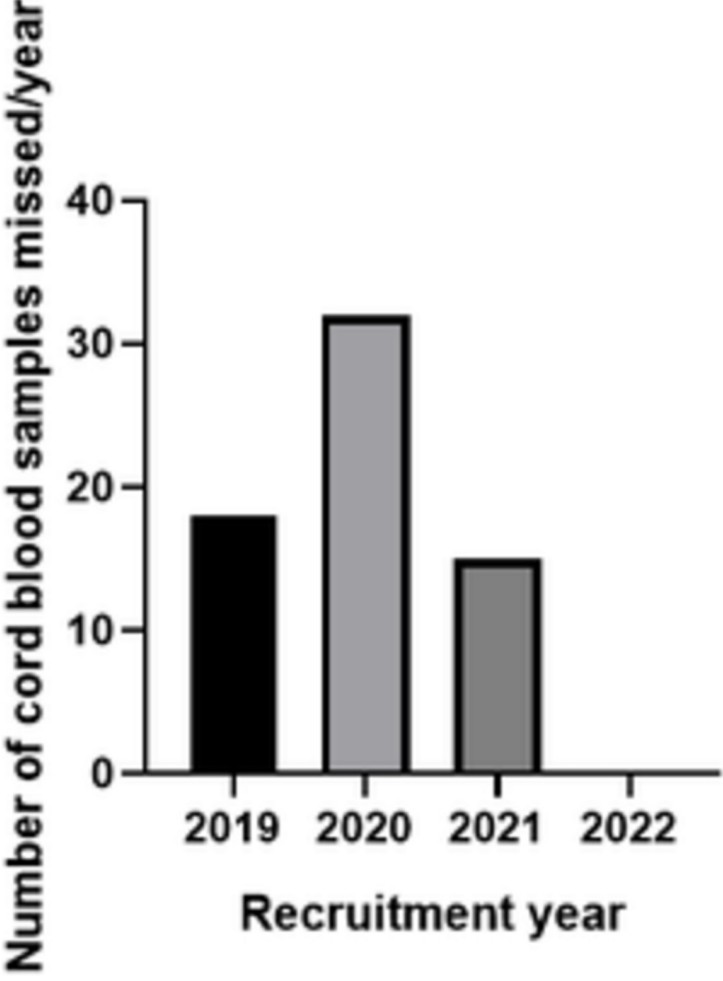

**Fig 3. Number of cord blood samples missed over the period of recruitment.**

## Acknowledgments

We are grateful to all research assistants, nurses and laboratory technicians for their contribution to the study, especially Portia Antwi, Richard Darko, Nancy Yeboah, Alberta Nimako, Theresa Quartey, Joshua Quarshie, Mario Rashid Kadiah and Millicent Arhen. We also would like to thank the patients for their help.

## Author Contributions

**Conceptualization:** Charlotte Osafo, Dwomoa Adu, Rulan S. Parekh.

**Data curation:** Charlotte Osafo, Nicholas Ekow Thomford, Abraham Carboo, Chris Guure.

**Formal analysis:** Charlotte Osafo, Nicholas Ekow Thomford, Chris Guure.

**Funding acquisition:** Charlotte Osafo, Rulan S. Parekh, David Burke.

**Investigation:** Charlotte Osafo, Nicholas Ekow Thomford, Jerry Coleman, Abraham Carboo.

**Methodology:** Charlotte Osafo, Nicholas Ekow Thomford, Jerry Coleman, Abraham Carboo, Perditer Okyere.

**Project administration:** Charlotte Osafo, Nicholas Ekow Thomford, Jerry Coleman, Perditer Okyere, Richard Adanu, David Burke.

**Resources:** Charlotte Osafo, Nicholas Ekow Thomford, Jerry Coleman, Abraham Carboo, Perditer Okyere, Dwomoa Adu.

**Supervision:** Charlotte Osafo, Dwomoa Adu, Richard Adanu, Rulan S. Parekh, David Burke.

**Validation:** Nicholas Ekow Thomford, Abraham Carboo, Chris Guure.

**Visualization:** Chris Guure.

**Writing – original draft:** Charlotte Osafo, Nicholas Ekow Thomford.

**Writing – review & editing:** Charlotte Osafo, Nicholas Ekow Thomford, Jerry Coleman, Abraham Carboo, Perditer Okyere, Dwomoa Adu, Richard Adanu, Rulan S. Parekh, David Burke.

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
