## [Decision Letter · Decision Letter 0]

20 Jun 2022

PONE-D-22-07467APOL1 Genotype Associated Risk for Preeclampsia in African populations: Rationale and protocol design for studies in women of African ancestry in resource limited settingsPLOS ONE

Dear Dr. OSAFO,

Thank you for submitting your manuscript to PLOS ONE. After careful consideration, we feel that it has merit but does not fully meet PLOS ONE’s publication criteria as it currently stands. Therefore, we invite you to submit a revised version of the manuscript that addresses the points raised during the review process.

Reviewers' comments:

Reviewer's Responses to Questions

**Comments to the Author**

1. Does the manuscript provide a valid rationale for the proposed study, with clearly identified and justified research questions?

Reviewer #1: Yes

Reviewer #2: Yes

2. Is the protocol technically sound and planned in a manner that will lead to a meaningful outcome and allow testing the stated hypotheses?

Reviewer #1: No

Reviewer #2: Yes

3. Is the methodology feasible and described in sufficient detail to allow the work to be replicable?

Reviewer #1: No

Reviewer #2: Yes

4. Have the authors described where all data underlying the findings will be made available when the study is complete?

Reviewer #1: Yes

Reviewer #2: No

5. Is the manuscript presented in an intelligible fashion and written in standard English?

Reviewer #1: Yes

Reviewer #2: Yes

6. Review Comments to the Author

You may also provide optional suggestions and comments to authors that they might find helpful in planning their study.

Reviewer #1: General comments:

This is a well-powered study to assess the role of fetal and maternal APOL1 renal risk variants in the development of preeclampsia in a West African setting. The authors also will be measuring a number of biomarkers but make no mention on how these will be used in the analyses. The authors mention, but do not elaborate on a longitudinal component to assess maternal events in mothers experiencing preeclampsia but this is not developed in the manuscript. Critical details on biomarkers, placental histology, and definitions of outcomes and events are not provided. Any information about statistical tests would be helpful.

Abstract:

Methods: The study will not recruit all pregnant – there are exclusion criteria. Change all to only or just delete all/

Introduction:

Common variants in the apolipoprotein L-1 gene (APOL1) only found on African

chromosomes termed G1 and G2, are potent risk factors for a spectrum of kidney diseases (14) but very rare in European ancestry individuals. This sentence needs to be re-written, the haplotypes are termed G1 and G2—not the chromosomes.

There is a strong relationship between kidney function and hypertension (15, 16) and looking at the uniqueness of the APOL1 gene to people of African descent, an in-depth role of this gene in preeclampsia in indigenous African women will be key to understanding preeclampsia genetics and biomarker innovations. Uniqueness seems like the wrong word here and sentence construction is awkward—try rewriting. I would suggest replacing with “and considering the importance of the APOL1 gene to people of African descent,…” and adding …and advance biomarker innovations.

Define high-risk genotype at first usage.

Among populations from Nigeria and Ghana, the APOL1 high-risk genotype frequency approaches 13% among those with kidney disease. What is the overall frequency of high-risk genotypes?

It is known that preeclampsia results in part from microangiopathy in the glomerulus of the kidney suggesting a potentially important role of APOL1in preeclampsia (19, 20). This statement should be supported by briefly stating the evidence that APOL1 is associated with (micro)angiopathy. There is other evidence suggesting a role for APOL1 in preeclampsia: mouse model with APOL1 placental expression with preclampsia and fetal wasting, higher circulating antibodies against APOL1 in preclampsia mothers.

The studies did not confer a 2-fold risk for preclampsia, APOL1 did. Please revise sentence.

Most studies have been African Americans and sampling has been mostly underpowered.

This statement should be toned down. Strengths of the two studies is that one had a replication cohort, which showed similar strong effect sizes and was powered at nearly 80% to detect OR>1.8 and the second was well powered. In aggregate these studies provide convincing evidence that APOL1 risk alleles increase risk of preeclampsia in African Americans. This study should allow a more precise OR and give some idea of the penetrance in African populations experiencing different demographics and environmental stressors than African descendants living in the USA.

Study Details

Aims and objectives:

Aim 2 is to determine perinatal outcomes, but elsewhere it is hinted that there will be 3 yrs follow-up of mothers to assess longitudinal outcomes—but this not mentioned. What is the aim and hypothesis and outcomes/events that will be measured for the follow-up?

Study design/population

The authors state that the main recruiting facility is in Ghana—are there others and have they been IRB approved? Or do they mean the only facility is at the KMTH?

Inclusion and exclusion criteria (Table 1): what is the justification for excluding women with a prior history—most of the participants are multiparious. Excluding mothers with prior history might under-estimate the true effect size of APOL1 risk genotypes.

Ethnicity:

5 ethnicities are listed, all with the potential of having different APOL1 allele frequencies. Have the investigators ascertained the allele frequencies for these different populations? How will population stratification be handled in the analyses? For example the high risk allele frequency among Hausa is ~7% , Akan 42%, Ga 21% and Ewe 28%.

Sample handling:

This section is less about sample handling and more about technology for documentation. Are the study participants the same as enrolled in an earlier study? Why given identifiers from an earlier study, presumably with the same investigators. It would make more sense to give each participant a unique identifier which identified both the study and the participant to avoid mix-ups in freezers or over time.

Will samples be barcoded—essential to reduce sample id errors. There is considerable detail about pitfalls and solutions to recruitment and sample collection, but not much detail of how samples will be stored, QC measures, etc.

Biomarkers:

More details about biomarkers—what are the specific biomarkers that will be quantified? At what time points. This protocol design will be used as a reference for the study design and protocol for ancillary studies in the future—more detail is needed. Details of study could go into supplementary data. Where will testing for these biomarkers be done?

Quality control:

Will creatinine and albumin be assessed at multiple time points? How will QC be implemented; how will consistency between labs and with labs over time be ascertained? How will CKD or other outcomes be defined for mother follow-up studies? Will two uACR and creatinine levels be collected at least 3 months apart? Will random duplicate samples be collected to validate laboratory consistency over time?

The section on pitfalls on recruitment and how they were overcome is useful, but could be shortened. More information on laboratory and histology would be welcome. For example, what section of placenta will be collected, how will this be standardized across time, and is there a pathologist specializing in placental pathology on the team.

Expand on the longitudinal arm of the study—what incident outcomes/events will be captured? Will all mothers be followed? Is this a stated aim of the study? What analyses will be performed?

What platform will be used to call APOL1 alleles. Will the genotyping be performed in Ghana or the USA? What is the capacity building plan for this study in Ghana?

Overall: This is a compelling and important study, but this reviewer finds the rationale and protocol design to be overly descriptive and short on technical details, particularly for details about timing of sample collection for biomarkers, laboratory protocols, and QC protocols, which would be helpful to investigators interested in ancillary studies or citing this study. A table listing the biomarkers for preeclampsia, kidney function, inflammation and time points for collection would be extremely helpful.

Reviewer #2: 1. Page 11 (numbers indicate overall page number in the pdf): Please clarify whether the study included early onset vs late onset preeclampsia – were both groups approached for enrollment?

2. Page 11: c/c study design of 700 cases and 700 controls. this should be clarified as the targe number, and an presentation of the interim enrollment number. It should be clarified that this was 1400 is the goal and that they only enrolled sl more than half of each intended group at the current time

Pages 12/13- refer to figures which are not included

Page 13: in what % of deliveries wase cord blood unavailable/not obtained? Similarly for placental tissue- in what percent of their current sample was it available.

Page 13: what are ‘endothelial markers for mother and child’?

Table 2: clarify what is meant by the titles: @20 and @30 weeks. Is this time of enrollment, time of onset of dx, time of delivery? this is unclear

similarly in table 2: please clarify what is meant by booking vs diagnosis (table 2)

Table2: lots of missing data – explain why this is the case?

there are some patients with a prior hx of PE when first onset was the criteria? What do the authors plan to do with them?

What birth defects were included?

Page 18: what do they mean by 1033 ‘data collection tools; have been obtained? They probably mean values, not tools.

Page 19: at the beginning of recruitment patients were ..recruited early in their pregnancy – but then they modified the recruitment strategy to make sure the time between recruitment and delivery did not take too long.The authors should discuss the implications of this approach, as I believe it will weight the cohort more towards late-onset preeclampsia

7. PLOS authors have the option to publish the peer review history of their article (what does this mean?). If published, this will include your full peer review and any attached files.

**Do you want your identity to be public for this peer review?** For information about this choice, including consent withdrawal, please see our Privacy Policy.

Reviewer #1: No

Reviewer #2: No

We look forward to receiving your revised manuscript.

Kind regards,

Vicente Sperb Antonello, MD, MSc, Phd

Academic Editor

PLOS ONE

Journal Requirements:

This project was supported by the Fogarty International Center of the National Institutes of Health

under Award Number K43TW011160. 

However, funding information should not appear in the Acknowledgments section or other areas of your manuscript. We will only publish funding information present in the Funding Statement section of the online submission form. 

CO- Fogarty International Center of the National Institutes of Health under Award Number K43TW011160.

The funders had and will not have a role in study design, data collection and analysis, decision to publish, or preparation of the manuscript.

---

## [Author Response · Author response to Decision Letter 0]

11 Sep 2022

Dear Dr. Antonello, 

Thank you for all the reviews, the input has been valuable in improving the manuscript.

Please find below our responses to the reviewers comments. Our responses are marked red in the document. 

Yours sincerely, 

Charlotte Osafo MBCHB, FWACP, FGCPS, MS

Authors Responses to Reviewers’ comments:

6. Review Comments to the Author

You may also provide optional suggestions and comments to authors that they might find helpful in planning their study.

Reviewer #1: General comments:

This is a well-powered study to assess the role of fetal and maternal APOL1 renal risk variants in the development of preeclampsia in a West African setting. The authors also will be measuring a number of biomarkers but make no mention on how these will be used in the analyses. The authors mention, but do not elaborate on a longitudinal component to assess maternal events in mothers experiencing preeclampsia but this is not developed in the manuscript. Critical details on biomarkers, placental histology, and definitions of outcomes and events are not provided. Any information about statistical tests would be helpful.

For the current study (K43 Career development award), funding was sought to explore the links between genetic variants and preeclampsia in people of West African descent. During the course of the study, we recognized the importance of establishing an accompanying biorepository, which will be available for future study in this well characterized cohort. We plan to develop a competitive RO1 grant application to measure biomarkers, conduct placental histology and also follow up both the babies and their mothers to determine the incidence of chronic kidney disease and other non-communicable diseases. For the biorepository, serum, urine, whole blood, as well as placenta samples are stored at -80 degrees Celsius for future biomarker measurement, placental histology etc. In this current study, we are funded to measure clinical laboratory tests including a full blood count, serum creatinine, urine albumin to creatinine ratio and liver function tests.

Abstract:

Methods: The study will not recruit all pregnant – there are exclusion criteria. Change all to only or just delete all/

We have deleted all

Introduction:

Common variants in the apolipoprotein L-1 gene (APOL1) only found on African

chromosomes termed G1 and G2, are potent risk factors for a spectrum of kidney diseases (14) but very rare in European ancestry individuals. This sentence needs to be re-written, the haplotypes are termed G1 and G2—not the chromosomes.

We have rewritten the sentence to capture haplotypes

There is a strong relationship between kidney function and hypertension (15, 16) and looking at the uniqueness of the APOL1 gene to people of African descent, an in-depth role of this gene in preeclampsia in indigenous African women will be key to understanding preeclampsia genetics and biomarker innovations. Uniqueness seems like the wrong word here and sentence construction is awkward—try rewriting. I would suggest replacing with “and considering the importance of the APOL1 gene to people of African descent,…” and adding …and advance biomarker innovations.

We have undertaken the necessary changes as requested by the reviewer. 

Define high-risk genotype at first usage.

High-risk genotypes were defined as requested.

Among populations from Nigeria and Ghana, the APOL1 high-risk genotype frequency approaches 13% among those with kidney disease. What is the overall frequency of high-risk genotypes?

It is known that preeclampsia results in part from microangiopathy in the glomerulus of the kidney suggesting a potentially important role of APOL1in preeclampsia (19, 20). This statement should be supported by briefly stating the evidence that APOL1 is associated with (micro)angiopathy. There is other evidence suggesting a role for APOL1 in preeclampsia: mouse model with APOL1 placental expression with preclampsia and fetal wasting, higher circulating antibodies against APOL1 in preclampsia mothers.

We have modified the statement to include the evidence

The studies did not confer a 2-fold risk for preclampsia, APOL1 did. Please revise sentence.

Sentence has been revised

Most studies have been African Americans and sampling has been mostly underpowered. This statement should be toned down.

The statement has been toned down 

Strengths of the two studies is that one had a replication cohort, which showed similar strong effect sizes and was powered at nearly 80% to detect OR>1.8 and the second was well powered. In aggregate these studies provide convincing evidence that APOL1 risk alleles increase risk of preeclampsia in African Americans. This study should allow a more precise OR and give some idea of the penetrance in African populations experiencing different demographics and environmental stressors than African descendants living in the USA.

We have amended the sentence

Study Details

Aims and objectives:

Aim 2 is to determine perinatal outcomes, but elsewhere it is hinted that there will be 3 yrs follow-up of mothers to assess longitudinal outcomes—but this not mentioned. What is the aim and hypothesis and outcomes/events that will be measured for the follow-up? We have added the proposed longitudinal follow up as aim 3

Study design/population

The authors state that the main recruiting facility is in Ghana—are there others and have they been IRB approved? Or do they mean the only facility is at the KMTH? 

Korle Bu Teaching Hospital is the largest referral center in Ghana and it is the only recruiting facility. However, the IRB obtained has federal wide coverage and can be leveraged to recruit at other facilities in Ghana if the need arises. 

Inclusion and exclusion criteria (Table 1): what is the justification for excluding women with a prior history—most of the participants are multiparious. Excluding mothers with prior history might under-estimate the true effect size of APOL1 risk genotypes.

 In addition to genetic risk, there are a number of other confounding factors such as socioeconomic status, concomitant hypertension, etc. If we include women who have history of preeclampsia, we are concerned about the need to assess for risk for recurrent preeclampsia with additional confounding versus initial episode of preeclampsia. We could conduct stratified analyses, however, we recognized that there was limited funding to adequately power each group. Hence, we simplified the study approach to first episode of preeclampsia.

Ethnicity:

5 ethnicities are listed, all with the potential of having different APOL1 allele frequencies. Have the investigators ascertained the allele frequencies for these different populations? How will population stratification be handled in the analyses? Population stratification will be handled through individual analysis to take out confounding effects and use pooled data. For example the high risk allele frequency among Hausa is ~7% , Akan 42%, Ga 21% and Ewe 28%. A recent study has should genotype distribution among some ethnic groups. Blazer A, Dey ID, Nwaukoni J, Reynolds M, Ankrah F, Algasas H, Ahmed T, Divers J. Apolipoprotein L1 risk genotypes in Ghanaian patients with systemic lupus erythematosus: a prospective cohort study. Lupus Sci Med. 2021 Jan;8(1):e000460. doi: 10.1136/lupus-2020-000460. PMID: 33461980; PMCID: PMC7816898

Sample handling:

This section is less about sample handling and more about technology for documentation.

We have put all the subheadings together under Recruitment Strategy

 Are the study participants the same as enrolled in an earlier study? 

The same investigators undertook an initial pilot study where 100 participants were enrolled. These participants will be added to the 1400 who are being recruited 

Why given identifiers from an earlier study, presumably with the same investigators. 

The identifiers are a continuation from the initial 100 individuals who were recruited into the pilot study

It would make more sense to give each participant a unique identifier which identified both the study and the participant to avoid mix-ups in freezers or over time. 

Each participant has been given a unique identifier together with their respective babies for easy identification. There is barcoding of all samples with dedicated freezer space

Will samples be barcoded—essential to reduce sample id errors. There is considerable detail about pitfalls and solutions to recruitment and sample collection, but not much detail of how samples will be stored, QC measures, etc.

Samples are given unique barcodes for easy identification.

Biomarkers:

More details about biomarkers—what are the specific biomarkers that will be quantified? At what time points. This protocol design will be used as a reference for the study design and protocol for ancillary studies in the future—more detail is needed. Details of study could go into supplementary data. Where will testing for these biomarkers be done?

Please refer to response comments . We have added table 2 which has the biomarkers that will be measured in future studies when funding is available. In this present study, we are not funded to measure preeclampsia biomarkers. When funding is available, these biomarkers will be measured in a research facility available to the research team. The two facilities are Noguchi Memorial Institute for Medical Research and MDS Lancet Laboratory. 

Quality control:

Will creatinine and albumin be assessed at multiple time points? How will QC be implemented; how will consistency between labs and with labs over time be ascertained? How will CKD or other outcomes be defined for mother follow-up studies? Will two uACR and creatinine levels be collected at least 3 months apart? Will random duplicate samples be collected to validate laboratory consistency over time? 

For the longitudinal study (which will be undertaken when funding is available), creatinine and UACR will be assessed annually at the MDS Lancet Laboratory. 

Random duplicated samples will be collected to validate the measurements and/or specimens will be divided for replication analyses of any biomarkers

.

The section on pitfalls on recruitment and how they were overcome is useful, but could be shortened. More information on laboratory and histology would be welcome. For example, what section of placenta will be collected, how will this be standardized across time, and is there a pathologist specializing in placental pathology on the team.

The section has been shortened. Section of placenta that will be collected has been added as requested. Research assistants follow a standardized protocol for collecting placental samples, which was added. There is no pathologist included in the current study. However, a placental pathologist will be included in the next phase of the project when funds are available. 

Expand on the longitudinal arm of the study—what incident outcomes/events will be captured? Will all mothers be followed? Is this a stated aim of the study? What analyses will be performed?

The longitudinal arm of the study is yet to be developed. We plan to follow up all mothers and their babies. Please refer to response under general comments

What platform will be used to call APOL1 alleles. Will the genotyping be performed in Ghana or the USA? What is the capacity building plan for this study in Ghana?

We have added a statement of where and how genotyping will be achieved. The TaqMan assay is a predesigned genotyping method that contain target specific primers and probes that are linked to the various alleles. These probes (one with a FAM dye label and one with a VIC dye label ) which emit a flourescent signal. A qPCR thermal cycler which in our case will be the CFX-96 thermal cycler will be programmed to read the signal. The genotypes come out as graphical output after using allelic discrimination as call method. Homozygous allele will all be on one side for wild type and mutant while heterozygous alleles will align in the centre. The data can be exported to excel for further analysis. There are QC checks for this procedure so it is robust and validation can be undertaken with Sanger sequencing. 

Overall: This is a compelling and important study, but this reviewer finds the rationale and protocol design to be overly descriptive and short on technical details, particularly for details about timing of sample collection for biomarkers, laboratory protocols, and QC protocols, which would be helpful to investigators interested in ancillary studies or citing this study. A table listing the biomarkers for preeclampsia, kidney function, inflammation and time points for collection would be extremely helpful.

We have added Table 2 which summarizes the biomarkers with time points for collection to help easy understanding

Reviewer #2: 1. Page 11 (numbers indicate overall page number in the pdf): Please clarify whether the study included early onset vs late onset preeclampsia – were both groups approached for enrollment?

Both early onset and late onset groups were initially approached for enrollment. However due to difficulty in following up the early onset cases, we focused more on the late onset group. 

2. Page 11: c/c study design of 700 cases and 700 controls. this should be clarified as the targe number, and an presentation of the interim enrollment number. It should be clarified that this was 1400 is the goal and that they only enrolled sl more than half of each intended group at the current time

Pages 12/13- refer to figures which are not included

We have deleted the Fig 5 from the statement

Page 13: in what % of deliveries wase cord blood unavailable/not obtained? Similarly for placental tissue- in what percent of their current sample was it available.

The actual numbers have been provided in the Figure 3 for cord blood samples missed which commensurate with the placenta samples. We have added a statement on the percentages.

Page 13: what are ‘endothelial markers for mother and child’?

Endothelial markers include tissue plasminogen activator, von Willebrand factor activity and antigen. However, we have deleted the “ endothelial markers for mother and child” since we do not have funding to measure that in the current study.

Table 2: clarify what is meant by the titles: @20 and @30 weeks. Is this time of enrollment, time of onset of dx, time of delivery? this is unclear

These are the gestational ages at which the measurements were undertaken, which is also the time of enrollment. 

similarly in table 2: please clarify what is meant by booking vs diagnosis (table 2)

Booking’ refers to the time the pregnant woman first reported for antenatal care . 

'Diagnosis’ is the time the pregnant woman was diagnosed with preeclampsia or the time the normotensive pregnant woman was recruited as control into the study.

Table2: lots of missing data – explain why this is the case?

The missing data has to do with the data collectors who had not uploaded the data to the RedCap server as at the time of writing this manuscript. Also some of the folders were not available to extract the data required

there are some patients with a prior hx of PE when first onset was the criteria? What do the authors plan to do with them?

The data will be stratified during analysis and controlled for as part of analysis. 

What birth defects were included?

These were generalised birth defects such as Congenital Heart Disease, chromosomal abnormalities, neural tube defects. 

Page 18: what do they mean by 1033 ‘data collection tools; have been obtained? They probably mean values, not tools.

This means the data collection questionnaire was made up of 1033 itemised questions or values

Page 19: at the beginning of recruitment patients were ..recruited early in their pregnancy – but then they modified the recruitment strategy to make sure the time between recruitment and delivery did not take too long.The authors should discuss the implications of this approach, as I believe it will weight the cohort more towards late-onset preeclampsia

This is true. It will weigh the cohort toward late onset however, the medical history of the patient will still inform us on the early onset of the patient we have recruited. WE may likely have two groups of early onset and late onset pre-eclamptic patients. 

7. PLOS authors have the option to publish the peer review history of their article (what does this mean?). If published, this will include your full peer review and any attached files.

Do you want your identity to be public for this peer review? For information about this choice, including consent withdrawal, please see our Privacy Policy.

Reviewer #1: No

Reviewer #2: No

Please submit your revised manuscript by Aug 04 2022 11:59P

---

## [Decision Letter · Decision Letter 1]

17 Oct 2022

PONE-D-22-07467R1APOL1 Genotype Associated Risk for Preeclampsia in African populations: Rationale and protocol design for studies in women of African ancestry in resource limited settingsPLOS ONE

Dear Dr. OSAFO,

Thank you for submitting your manuscript to PLOS ONE. After careful consideration, we feel that it has merit but does not fully meet PLOS ONE’s publication criteria as it currently stands. Therefore, we invite you to submit a revised version of the manuscript that addresses the points raised during the review process.

We look forward to receiving your revised manuscript.

Kind regards,

Vicente Sperb Antonello, MD, MSc, Phd

Academic Editor

PLOS ONE

Journal Requirements:

Additional Editor Comments:

APOL1 Genotype Associated Risk for Preeclampsia in African populations: Rationale and protocol design for studies in women of African ancestry in resource limited settings

After careful evaluation of the article and data from by reviewers, I understand that the article should pass through a minor revision to be accepted into Plos One. Please check the reviewers´ recommendations and adjust in the present article.

Reviewers' comments:

Reviewer's Responses to Questions

**Comments to the Author**

1. Does the manuscript provide a valid rationale for the proposed study, with clearly identified and justified research questions?

Reviewer #1: Yes

Reviewer #2: Yes

2. Is the protocol technically sound and planned in a manner that will lead to a meaningful outcome and allow testing the stated hypotheses?

Reviewer #1: Yes

Reviewer #2: Yes

3. Is the methodology feasible and described in sufficient detail to allow the work to be replicable?

Reviewer #1: Yes

Reviewer #2: Yes

4. Have the authors described where all data underlying the findings will be made available when the study is complete?

Reviewer #1: Yes

Reviewer #2: Yes

5. Is the manuscript presented in an intelligible fashion and written in standard English?

Reviewer #1: Yes

Reviewer #2: Yes

6. Review Comments to the Author

You may also provide optional suggestions and comments to authors that they might find helpful in planning their study.

Reviewer #1: The authors have addressed this reviewer's comments. I have only two minor edits: APOL1 when referring to the gene is always italicized; when referring to the protein it is not. Please correct in abstract and throughout text

Preeclampsia is a disease and is therefore not capitalized in abstract and elsewhere.

Reviewer #2: I believe the authors have addressed the reviewer's initial comments satisfactorily. There are two areas that could use some additional clarification (but not re-review). First, limiting the sample to women without a history of preeclampsia removes from the population women who might have recurrent preeclampsia due to their (or their fetus's) APOL1 status. This would result in an underestimate of the relationship between apol1 and preeclampsia generally, and is worthy of a statement to this regard in the discussion.

Second, there is still a little confusion in the manuscript about when the maternal blood draw occurs. According to table 2, biomarkers and blood samples are taken at recruitment, but I believe that they are actually taken at delivery (as stated in the section above entitled 'Sample collection and laboratory assays'. it would be good to go thru the manuscript one last time and clarify exactly when these samples are obtained.

7. PLOS authors have the option to publish the peer review history of their article (what does this mean?). If published, this will include your full peer review and any attached files.

Reviewer #1: No

Reviewer #2: No

---

## [Author Response · Author response to Decision Letter 1]

18 Oct 2022

October 18, 2022

Dear Dr. Antonello, 

Thank you for all the reviews, the input has been valuable in improving the manuscript.

Please find below our responses to the reviewers comments. Our responses are marked red in the document. 

Yours sincerely, 

Charlotte Osafo MBCHB, FWACP, FGCPS, MS 

Review Comments to the Author

Reviewer #1: The authors have addressed this reviewer's comments. I have only two minor edits: APOL1 when referring to the gene is always italicized; when referring to the protein it is not. Please correct in abstract and throughout text

We have italicised all APOL1 genes

Preeclampsia is a disease and is therefore not capitalized in abstract and elsewhere.

We have done as suggested. 

Reviewer #2: I believe the authors have addressed the reviewer's initial comments satisfactorily. There are two areas that could use some additional clarification (but not re-review). First, limiting the sample to women without a history of preeclampsia removes from the population women who might have recurrent preeclampsia due to their (or their fetus's) APOL1 status. This would result in an underestimate of the relationship between apol1 and preeclampsia generally, and is worthy of a statement to this regard in the discussion.

We have included a statement to that effect “This may result in an underestimate of the relationship between APOL1 and preeclampsia”. 

Second, there is still a little confusion in the manuscript about when the maternal blood draw occurs. According to table 2, biomarkers and blood samples are taken at recruitment, but I believe that they are actually taken at delivery (as stated in the section above entitled 'Sample collection and laboratory assays'. it would be good to go thru the manuscript one last time and clarify exactly when these samples are obtained.

We have taken a look and we are ok with the version displayed in the mansucript

---

## [Editor Report · Decision Letter 2]

10 Nov 2022

APOL1 Genotype Associated Risk for Preeclampsia in African populations: Rationale and protocol design for studies in women of African ancestry in resource limited settings

PONE-D-22-07467R2

Dear Dr. OSAFO,

We’re pleased to inform you that your manuscript has been judged scientifically suitable for publication and will be formally accepted for publication once it meets all outstanding technical requirements.

Kind regards,

Vicente Sperb Antonello, MD, MSc, Phd

Academic Editor

PLOS ONE

Additional Editor Comments (optional):

APOL1 Genotype Associated Risk for Preeclampsia in African populations: Rationale and protocol design for studies in women of African ancestry in resource limited settings

Regarding the present manuscript, all reviewers' notes have been clarified and I believe that this article should be accepted for publication in Plos One.

---

## [Editor Report · Acceptance letter]

14 Nov 2022

PONE-D-22-07467R2 

*APOL1* genotype associated risk for preeclampsia in African populations: Rationale and protocol design for studies in women of African ancestry in resource limited settings 

Dear Dr. Osafo:

I'm pleased to inform you that your manuscript has been deemed suitable for publication in PLOS ONE. Congratulations! Your manuscript is now with our production department. 

Kind regards, 

on behalf of

Dr. Vicente Sperb Antonello 

Academic Editor

PLOS ONE